# Design Strategies and Precautions for Using Vaccinia Virus in Tumor Virotherapy

**DOI:** 10.3390/vaccines10091552

**Published:** 2022-09-17

**Authors:** Xinjun Liu, Jian Zhao, Xiaopeng Li, Fengxue Lao, Min Fang

**Affiliations:** 1CAS Key Laboratory of Pathogenic Microbiology and Immunology, Institute of Microbiology, Chinese Academy of Sciences, Beijing 100101, China; 2University of Chinese Academy of Sciences, Beijing 101408, China; 3School of Life Sciences, Henan University, 85 Minglun Street, Kaifeng 475001, China; 4Key Laboratory for Major Obstetric Diseases of Guangdong Province, the Third Affiliated Hospital of Guangzhou Medical University, Guangzhou 524023, China; 5Beijing Key Laboratory of Bioactive Substances and Functional Foods, Beijing Union University, Beijing 100191, China; 6International College, University of Chinese Academy of Sciences, Beijing 101408, China

**Keywords:** oncolytic virotherapy, vaccinia virus, tumor treatment, personalized treatment

## Abstract

Oncolytic virotherapy has emerged as a novel form of cancer immunotherapy. Oncolytic viruses (OVs) can directly infect and lyse the tumor cells, and modulate the beneficial immune microenvironment. Vaccinia virus (VACV) is a promising oncolytic vector because of its high safety, easy gene editing, and tumor intrinsic selectivity. To further improve the safety, tumor-targeting ability, and OV-induced cancer-specific immune activation, various approaches have been used to modify OVs. The recombinant oncolytic VACVs with deleting viral virulence factors and/or arming various therapeutic genes have displayed better therapeutic effects in multiple tumor models. Moreover, the combination of OVs with other cancer immunotherapeutic approaches, such as immune checkpoint inhibitors and CAR-T cells, has the potential to improve the outcome in cancer patients. This will open up new possibilities for the application of OVs in cancer treatment, especially for personalized cancer therapies.

## 1. Introduction

OVs are a natural or genetically-modified class of viruses able to selectively infect and kill cancer cells. In addition to the oncolytic activities, OVs also can stimulate cancer-specific immune responses [1,2]. At present, many viruses, such as herpes simplex virus (HSV), adenoviruses, measles virus, poxviruses, reovirus, and Newcastle disease virus (NDV) [3], have been used as OVs and are engineered to attenuate pathogenicity and enhance immunogenicity for achieving optimal tumor treatment. Some are already approved by FDA, for example, talimogene laherparepvec (T-VEC), a modified herpes simplex virus (HSV) for melanoma treatment [4]. Some OVs have entered clinical trials, such as HSV G207 and telomerase-specific adenovirus OBP-301 (Telomelysin) [5], used for the treatment of recurrent high-grade glioma (in phase II clinical trial) (NCT04482933) and unresectable metastatic melanoma (in phase II clinical trial) (NCT03190824), respectively. VACV has also become one of the most successful vectors for tumor treatment [6]. Oncolytic VACVs have shown good application prospects in clinical trials of many cancers, such as melanoma, colorectal cancer, and pediatric cancer [7,8,9,10].

## 2. Oncolytic Vaccinia Vectors and Oncolytic Effects

VACV, a member of the *orthopoxvirus* of *poxviridae*, is a linear double-stranded DNA virus [11]. VACV has been used as a live-attenuated vaccine to eradicate smallpox, and the safety of VACV in humans has been proved by the practice. Moreover, immunization against smallpox was discontinued in 1980. Currently, a large proportion of people have not been exposed to VACV, which is also good merit for using OVs in cancer patients. According to the characteristics, pathogenicity, and host range, VACV can be divided into a variety of strains, including the New York City Board of Health (NYCBH) strain, Western Reserve (WR) strain, Wyeth strain, Modified Vaccinia Ankara (MVA) strain, Copenhagen strain, Tian Tan strain, etc. [11,12]. VACV can directly infect and lyse tumor cells. Furthermore, VACV infection can promote local and systemic antitumor immunity to indirectly control tumor growth [13]. After tumor cells lysis, the viral particles released from tumor cells whereafter infect adjacent or distant tumor cells for further tumor destruction. Meanwhile, an enhanced immune response can be elicited due to the released pathogen-associated molecular patterns (PAMPs), danger-associated molecular pattern signals (DAMPs), and some tumor-associated antigens (TAAs), especially neoantigens, which may overcome the local immunosuppressed tumor microenvironment and even create an in situ immunization effect through cross-presenting TAAs to host immune system [10,14].

As an oncolytic vector, VACV has several advantages [10]. Firstly, VACV has a rapid oncolytic replication cycle. Studies have shown that VACV produced 4–6 logs more viruses than adenovirus at 48 h post-infection, and the caused cytopathic effects were also greater at lower MOI and at earlier time points than adenovirus [15]. Secondly, the use of VACV is generally safe and reliable. It has been widely used in humans during the smallpox eradication campaign, and if adverse reactions occur, anti-viral sera or immunoglobulin and some drugs that have been approved or proved to have antiviral effects, such as cidofovir or ST-246, can be used to treat viral infection [12,16]. Thirdly, VACV has a large genome of about 190 kb in which foreign DNA up to 25kb can be inserted without deleting any viral sequences [11], making it easier for genetic engineering and targeted modification. In addition, VACV has intrinsic selectivity for tumor cells. It was found that VACV strains have a higher replication level in tumor cell lines than in normal cells when using a series of VACV strains including WR and Wyeth to infect normal and tumor cells [17].

## 3. Strategies and Approaches to Enhance the Anti-Tumor Capacity of VACV

To further improve the tumor selectivity and antitumor efficacy of VACV, various modified approaches have been used, including deletions of viral virulence genes such as thymidine kinase (TK) and vaccinia growth factor (VGF), insertions of therapeutic genes such as granulocyte-macrophage colony-stimulating factor (GM-CSF), co-stimulating factors and bispecific T cell engagers (BiTEs), and insertions of TAA especially neoantigens, etc., which enhance the antitumor capacity of VACV.

### 3.1. Vaccinia Virus Virulence Genes

In normal cells, a range of genetic products produced by wild type VACV can render cells to adapt to viral reproductions, including thymidine kinase (TK) and viral growth factor (VGF) [12,18]. TK is one of the key enzymes in DNA metabolism [19]. In the replication process of VACV, a high concentration of nucleic acid pool generated with TK is needed to complete the replications of virus progeny [20]. When the TK gene is deleted, the replication of VACV will be restricted and display obvious tumor selectivity since there is usually a high activity of TK in tumor cells [21,22]. Therefore, TK gene deletion becomes an effective method to improve tumor selectivity in oncolytic VACV therapy. Comparing the replication ability between wild type and TK gene knocking-down Wyeth strains VACV, both viruses replicated effectively in human PANC-1 pancreatic cancer cells and human MCF-7 breast cancer cells, and viral titers were almost the same. However, in non-proliferating or non-transformed cells, the replication capacity of TK knocking-down particles was significantly reduced about 10 times [22], indicating the preference to tumor cells of VACV after TK gene deletion. JX-594, a modified Wyeth strain VACV through insertion of GM-CSF into the TK gene for TK deletion, also displayed an obvious selectivity in tumor cells [23], further demonstrating an advantage of TK gene deletion in tumor selectivity.

VGF is a homologous gene to epidermal growth factor (EGF). It has an EGF-like activity in virus-infected cultures, which can compete with EGF for binding to EGF membrane receptors (EGFR) to mediate localized cellular proliferation [24]. VACV often promotes the proliferation of host and peripheral cells through VGF to achieve its own replication and proliferation [12], while the absence of that factor significantly reduces the viral virulence and damage to normal cells [25]. Thus, VGF-deleted VACV has also been broadly applied in tumor therapy. In proliferating Swiss 3T3 cells, the VGF gene is not necessary for VACV replication. However, in stationary and contact-inhibited Swiss 3T3 cells, the replication efficiency and lethality of VGF-deficient VACV are reduced compared with wide type VACV [25], indicating the enhanced tumor selectivity after VGF deletion. To further achieve tumor selectivity, TK and VGF genes were both deleted to form a double-deleted VACV (vvDD). vvDD displayed comparable replication ability in dividing cells as TK or VGF single knockout virus. However, in resting cells, the replication ability of vvDD was significantly reduced than TK or VGF single knockout VACV [18]. Thus, vvDD can be used as a good oncolytic vector for further enhancement of tumor selectivity. JX-963, formed by vvDD with the insertion of GM-CSF, shows a greater efficacy against the primary and metastasizing lung tumors either compared with vvDD or JX-594 (Wyeth single TK-deleted VACV expressing GM-CSF) both in vitro and in vivo [17].

In addition, A48R and B18R are also deleted for tumor selectivity [26]. A48R codes for thymidylate kinase which participates in nucleotide metabolism and the deletion of A48R attenuates the WR virus. The B18R gene codes for a soluble type I interferon (IFN) receptor protein, which interferes with anti-viral responses to protect from the antiviral effects of type I IFN [27]. Unlike normal cells, tumor cells usually show IFN deficiency [28]. Thus, VACV lacking the B18R gene also reduces virulence and tends to replicate in tumor cells [26]. C7L and K1L are two important genes in regulating the virus-host range, including the infection capability of VACV in human cells. Deletions of C7L and K1L lead to deficient replication capability in mammalian cells and decreased virulence in mice. The deletions of C7L and K1L are also used for VACV attenuation to create a more safe and more effective HIV vaccine vector [29].

### 3.2. Cytokines and Chemokines

#### 3.2.1. Cytokines

Cytokines are a class of proteins secreted by immune cells and tissue cells, which can regulate the immune response by binding to corresponding receptors. Cytokine therapy for tumor patients has been an important treatment modality. However, cytokine in monotherapy does not fully achieve the efficacy seen in preclinical trials [30], as it is often associated with severe dose-limiting toxicities because of its short half-life [30,31]. Therefore, new strategies to avoid its disadvantages for tumor treatment are needed. Reports showed that utilizing VACV to deliver cytokines not only makes cytokines persist but also enhances the antitumor immunity induced by VACV [17,32,33].

GM-CSF is associated with a variety of immune cell activation and differentiation. GM-CSF can increase the expression of class Ⅱ major histocompatibility complex (MHC) molecules on the surface of macrophages [7,34], and recruit and activate dendritic cells (DC). When mice bearing established Lewis lung carcinoma were pretreated with docetaxel followed by GM-CSF-producing tumor vaccine, significant tumor regression, and prolonged survival were observed compared with chemotherapy alone, further studies revealed that augmentation of vaccine-induced antitumor immunity in docetaxel-treated mice primarily due to the enhanced survival of antigen-experienced T cells [35]. OVs generated with GM-CSF expression include adenovirus [36,37], herpes viruses [38], and VACV, all display positive tumor treatment outcomes. Recombinant VACV expressing GM-CSF, such as JX-594 and JX-963 [17,22,39,40], display potential therapeutic effects with safety and enhanced antitumor efficacy, including the destruction of tumor blood vessels [39,41]. JX-594 has shown exciting results in phase I clinical trials targeting a variety of tumors as local or systemic therapy, and a phase Ⅲ clinical trial has also recently been completed targeting hepatocellular carcinoma (Table 1).

Other cytokines, such as interferons (IFN) and interleukins (IL), have also displayed promising results in oncolytic viral therapy. IFNs have multiple antitumor mechanisms, including direct inhibition of tumor cell proliferation, anti-angiogenesis, and induction of tumor-specific cytotoxic T-cells [45]. IFN-β expressing oncolytic VACV (*TK^−^/B18R^−^/IFN-β^+^*, JX-795) therapy significantly increased the survival of breast cancer-bearing mouse models compared to the control group [32,46]. IL-12 plays a key role in augmenting the cytotoxic activity of T cells and natural killer cells. A recombinant VACV vKT0334 containing the genes encoding murine IL-12 efficiently infected a variety of tumor cell lines and produced high amounts of biologically active mIL-12, and the antitumor activity of vKT0334 was also significantly enhanced compared with the backbone virus (vKT033, VACV do not express IL-12) in sarcoma tumor mice model [33]. Other interleukins, such as IL-15, also display beneficial antitumor effects after cooperating with VACV vector. A recombinant VACV expressing a superagonist IL-15 (a fusion protein of IL-15 and IL-15Rα) led to significant tumor regression and extended survival of the mice bearing colon or ovarian cancer [47].

Although cytokine expression obtains good results, it is at the cost of reduced oncolytic activity due to the cytokine-associated immunity resulting in less viral replication and earlier clearance [48]. Toxicity effects have also been reported with systemic cytokine release caused by OVs-driven cytokine expression [31,48,49]. Cytokines are potent and complex immune mediators, designing cytokine expression recombinant oncolytic virotherapy requires a profound knowledge of cytokine biology and contemporary biotechnology to exploit their anti-tumor activity while minimizing toxicity.

#### 3.2.2. Chemokines

Chemokines are secreted chemotactic cytokines that mediate the migration and positioning of immune cells within various tissues. Chemokines influence the host immune response to tumors by directing leukocyte trafficking into tumor lesions to mediate both antitumor and protumor immune effects [33]. Among the chemokines, the type-1 chemokines CXCL9, CXCL10, and CXCL11 contribute to the inhibition of angiogenesis and tumor progression [50,51]. They recruit effector CD8^+^ T cells and Th1 cells expressing CXC-chemokine receptor 3 (CXCR3), the common receptor of CXCL9, CXCL10, and CXCL11, to migrate into tumors for mediating antitumor immunity. Tumors with higher-level expressions of CXCL9, CXCL10, and CXCL11 displayed increased infiltration of antitumor T lymphocytes [32]. CXCL11-producing tumors were completely rejected in a T-cell lymphoma mice model [52]. VACV expressing CXCL11 increased CXCL11 expression levels in virus-infected cells and stimulated T cells trafficking to malignant tissues, resulting in prolonged survival of mice with TC1 murine lung tumors, and the proliferation ability of the recombinant virus was not affected by CXCL11 expression, demonstrating the good antitumor potential of oncolytic CXCL11 expressing VACV [32,53]. In another study, the expression of CXCL11 on VACV also showed an enhanced therapeutic efficacy without affecting virus infection and replication, displaying increased local numbers of CD8^+^ CTLs as well as reduced expression of several suppressive molecules such as TGF-β, COX2, and CCL22 in the tumor microenvironment (TME), thus prolonged the survival of mice with a murine AB12 mesothelioma model [54].

CCL5 is also a potent chemoattractant that can attract NK cells, T cells, and DC expressing its receptor CCR5 [55]. CCL5 plays a crucial role in maintaining several functions of T cells such as survival, migration, and differentiation [56]. Reports showed that CCL5-armed VACV (vvDD-CCL5) enhanced immune cell infiltration into colorectal tumors in mice. Moreover, CCL5 expression also resulted in prolonged persistence of the virus exclusively within the tumor. Further enhancement of antitumor effectiveness was achieved when vvCCL5 was used in conjunction with DC1 vaccination, correlating with increased immune cell infiltration into the tumor and an apparent Th1 skewing of the infiltrating T-cells [57].

Hot tumors are typically characterized by the accumulation of pro-inflammatory cytokines, CD8^+^ T cells, and NK cells; these malignancies normally have a better response rate to immunotherapy treatment. Conversely, cold malignancies present immunosuppressive immune cells in the tumor, including Tregs, and tend to have a poor prognosis and response to immune checkpoint inhibitors [58]. Utilizing VACV vectors to deliver chemokines or VACV combined with chemokine therapy may alter the TME, recruit CD8^+^ T cell and NK cells into the tumor, and turn the cold tumor into a hot tumor to achieve better therapeutic antitumor efficacy (Figure 1).

### 3.3. Costimulators

T cells play a key role in controlling tumor growth. There is growing evidence that effective T cell activation can control tumor growth and prolong the survival of tumor patients [59,60,61,62]. For T cell activation, the second or costimulatory signal between T cell and costimulating molecules on antigen-presenting cells (APC) is needed and necessary under natural physiological conditions [63,64]. B7-1 (CD80) and B7-2 (CD86) are two important costimulatory molecules presenting on the surface of specialized APC and interact with CD28 or cytotoxic T-lymphocyte antigen (CTLA-4) expressed on T cells [65]. CD28 delivers a positive stimulatory signal to T cells [66], while CTLA-4 delivers a negatively regulating signal to T cells [67]. The affinity of CTLA-4 to B7 is much higher than CD28, thus the expression of CTLA-4 dampens CD28 co-stimulating signal of T cells by competing with CD28 to bind to B7, which leads to B7 consumption and T cells suppression [68]. Some studies found that B7-1 is also expressed in some tumors, the expression of B7-1 in metastatic lesions is much lower than in normal tissues, implying the correlations of B7-1 downregulation with tumor progression [69,70]. Moreover, most mouse and human tumors do not express B7-1 or B7-2. Thus, even though a potent antigen is expressed on the tumor cells, it is unlikely to activate antitumor T-cell responses due to the absence of costimulators [63]. Therefore, increasing the expression of B7 molecules on tumors can provide co-stimulation to elicit effective antitumor T cell immunity.

VACV vector is a good tool to deliver B7 molecules, which can turn tumor cells themselves into APC to provide a costimulatory signal for T cells after infection, thus triggering potent tumor-specific T cell responses [71]. Mouse melanoma cells infected with a set of recombinant VACV encoding the murine B7-1 or/and B7-2 provided effective costimulation, resulting in the proliferation of resting CD4^+^ T cells. Moreover, when B7-1 and B7-2 were delivered together on the same cell, the proliferative response of CD4^+^ T cells was further increased [72]. In another study, inoculation of rV-B7-1- or rV-B7-2 (recombinant vaccinia viruses containing the murine B7-1 or B7-2 genes)-infected tumor cells into immunocompetent animals resulted in no tumor growth [63]. In a melanoma study, VACV vectors expressing B7-1 effectively overcame the immunosuppressive TME [67,73] and enhanced tumor-specific T cell responses. Therefore, the utility of recombinant vaccinia viruses to deliver B7 molecules to tumor cells is a potential strategy to enhance antitumor T cell responses.

### 3.4. BiTE

Bispecific antibodies are engineered proteins that can simultaneously engage with two different antigens [74]. Bispecific T cell engager (BiTE) is generated by single-chain variable fragments (scFv) of two different monoclonal antibodies, one of which is usually a CD3 agonist to engage with CD3 on T cells, and the other can recognize a surface antigen on target such as tumor cells. They are connected through a short, flexible peptide chain [75]. Through co-engaging with CD3 and targeted cell antigens, BiTE-mediated T cell activation can occur without co-stimulating signals or in vitro prestimulation, which is also independent of the presentation of antigen peptides on surface MHC molecules [76,77]. Therefore, tumor cell killing mediated by BiTE is no longer limited to the expression of MHC molecules on targeted cells, even if MHC expression has been lost by tumor cells, which is usually an immune evasion strategy of tumors [78].

Though the advantages of BiTE bring promising in solid tumor patients treatment, challenges are still faced in the application of BiTE against common cancers, such as variable mutational burden in tumors, poor BiTE drug delivery into tumors, an overwhelmingly immunosuppressive TME which may make BiTE hard to work, off-target toxicities against adjacent normal cells, dose-dependent severe systemic side effects, and so on [79].

Fibroblast activation protein (FAP) expression is closely related to the proliferation and apoptosis of tumor-related fibroblasts. Reports showed that the generated BiTE-VACV with an anti-mouse CD3 and an anti-FAP, mFAP-TEA-VV, mediated effective T cell killing of FAP expressing cells, which was accompanied by increased secretion of IFN-γ and IL-2. In vivo, mFAP-TEA-VV enhanced viral titer within the tumor and had potent antitumor activity in comparison to control VVs in an immunocompetent B16 melanoma mouse model [80]. EphA2-TEA-VV is a recombinant VV encoding a secretory bispecific T-cell engager consisting of two single-chain variable fragments specific for CD3 and the tumor cell surface antigen EphA2. Compared with the unmodified virus, the replication and oncolysis abilities of EphA2-TEA-VV were not changed. EphA2-TEA-VV not only killed infected tumor cells but also induced bystander killing of noninfected tumor cells. Furthermore, EphA2-TEA-VV had potent antitumor activity in comparison with control VV in a lung cancer xenograft model [59].

### 3.5. Tumor Antigens

Tumor antigens (TAs) can be generally categorized as tumor-associated antigens (TAAs) and tumor-specific antigens (TSAs) [81]. Though many studies have employed TAA in virotherapy, including oncolytic VACV [82,83,84,85,86], since TAAs express not only in tumor cells but also in healthy tissues [81,87], they may also lead to off-target effects. Therefore, targeting TSAs is a good way to treat tumors. TSAs are either oncogenic viral antigens or tumor-specific neoantigens [88]. Oncogenic viral antigens are identified in virus-induced tumors. Clinical antitumor efficacy of VACV vectors armed with oncogenic viral antigens is confirmed in oncogenic virus clearance and tumors regression, such as oncolytic VACV containing E6 and E7 antigens from human papillomavirus 16 (HPV16) in HPV eradication and against cervical cancer [89,90]. HER2 is overexpressed and/or amplified in several types of tumors, such as breast cancer and gastric cancer [91]. Previous studies demonstrated enhanced efficacy of oncolytic VACV recombinants encoding combined HER2 and GM-CSF in modulating the microenvironment of myeloid-derived suppressor cells (MDSCs)-rich tumors [92].

Tumor-specific neoantigens are generated by tumors through multiple mutation events [93,94]. High mutation burden neoantigens (high authentic neoantigens) might induce successful antitumor immunity as potent immunogens [95,96,97]. Many tumor vaccines are generated with neoantigens [98]. However, when the tumors are with intermediate to low mutation burdens, the application of neoantigen-targeted vaccines is highly limited since the detection of neoantigens is hard due to the technical limitations of detection [99]. Excitingly, with the development of deep-sequencing technologies, it becomes possible to identify the mutations and predict potential neoantigens [100]. Bioinformatic analysis of The Cancer Genome Atlas (TCGA) data revealed that increased point mutations and neoantigen burdens of tumors are correlated with increased cytotoxic T cells infiltration [99], implying that combining neoantigen with oncolytic VACV might increase the expression of low burden neoantigens in tumors, thus eliciting anti-tumor neoantigen-specific T cells, and expanding the antitumor efficacy. In addition, the anti-tumor immunity can be further amplified by the newly released neoantigens taken up by APC after tumor cell lysis, thus producing a more direct and effective antitumor immunity. Since the majority of neoantigens are exclusive to tumors [93,94] and mutated unique to individual patients [101], the therapy of oncolytic VACV armed with neoantigens also means personalized therapy [102]. Accordingly, oncolytic VACV armed with neoantigens is a personalized oncolytic agent, which can be tailor-made for tumor improving and curing. A simplified mechanism of neoantigen-armed oncolytic VACV is illustrated in Figure 2 [13,103].

### 3.6. Combinations

#### 3.6.1. Combination with Pharmaceutical Drugs

Oncolytic viral therapy faces the limitation of viral clearance due to the generation of neutralizing antibodies, the presence of Abs specific to VACV can prohibit the booster effect of recombinant VACV vectors [104]. Therefore, combination strategies are elicited to circumvent these problems, thus achieving a successful vaccination and even boosting the viral oncolytic effects.

Cyclooxygenase-2 (COX-2) is a key enzyme in fatty acid metabolism, which is upregulated during cancer. Enhanced COX-2 activity promotes angiogenesis, inhibits apoptosis, and increases the metastatic potential of carcinogenesis [105]. The COX-2 inhibitor is a class of nonsteroidal anti-inflammatory drugs. It inhibits the activity of the COX-2 enzyme and reduced antibody production specific to VACV by inhibiting antibody induction [106]. One study found that COX-2 inhibitors prevented the generation of VACV-specific neutralizing antibodies, thus circumventing the limitation of viral clearance without losing infectivity in mice vaccinated with VACV previously [107], indicating the assistant of COX-2 inhibitors in protecting VACV clearance. Further, the treatment of VACV in combination with COX-2 inhibitors regressed ovarian tumors and enhanced the survival of tumor-bearing mice, the efficacy was more effective than either administration of VACV or COX-2 inhibitors alone [107]. The product of COX-2 activity, prostaglandin E2 (PGE-2), is a key mediator of resistance to immunotherapies [108]. The elevation expression of PGE-2 couples with an increase in the suppressive chemokine profile and a high level of granulocytic myeloid-derived suppressor cells (MDSC), which can result in a loss of immunotherapeutic potential [109]. Therefore, COX-2 inhibitors can block PGE-2 activity and, thus, reverse the PGE-2-created tumor immunosuppressive microenvironment, synergizing with oncolytic VACV for TME remolding [110], further confirming the benefit of COX-2 inhibitors and oncolytic VACV in combinations against tumors. Indeed, studies showed that vaccinia vectors engineered to target PGE-2 overcame localized immunosuppression and sensitized established tumors resistant to immunotherapy [109].

Sunitinib is a receptor tyrosine kinase inhibitor. It has also been used to combine with VACV to assist the antitumor activity as sunitinib can decrease Treg and MDSC in some tumors [111,112,113]. In a carcinoembryonic antigen transgenic (CEA-Tg) mice model, the combined treatment of sunitinib and modified VACV reduced tumor volumes and increased survival of CEA-Tg mice, concomitantly with increased intratumoral antigen-specific T lymphocytes infiltration and decreased immunosuppressive immune cells [114]. In a renal tumor mice model, oncolytic VACV in combination with sunitinib also significantly increased the overall survival benefit of tumor mice more than VACV or sunitinib treatment alone [115]. In addition, other pharmaceutical drugs, such as histone deacetylase inhibitor trichostatin A [116], chemokine modulating drug cocktail [117], and deoxyadenosine analog cytarabine [118] are also employed to combine with oncolytic VACV to explore better tumor treatment.

#### 3.6.2. Combination with Anti-Angiogenesis Strategies

Angiogenesis is an important part of tumor growth and development [119]. In the period of malignancy, tumor cells start to secrete angiogenic growth factors which bind to receptors on endothelial cells of nearby blood vessels, initiating new vessel formation to support further tumor growth [119]. Factors, such as vascular endothelial growth factor (VEGF) and epidermal growth factor receptor (EGFR), play crucial roles in tumor angiogenesis to support tumor initiation and development [120], and the expression levels of VEGF and EGFR are highly correlated with the incidence of tumor growth and poor prognosis [121,122,123,124,125,126]. Thus, VEGF and EGFR have been proposed as key targets to combine with oncolytic viral therapy for exploring a better regimen against tumors.

Oncolytic VACV has been shown to induce a profound, rapid, and tumor-specific vascular collapse in both preclinical models and clinical studies. It was observed that revascularization after viral therapy was dramatically delayed and did not occur until after viral clearance, and VEGF levels in the tumor were suppressed throughout the period of active viral infection [41,115,127]. In other studies, VACV vectors armed with single-chain antibodies against VEGF also presented a significant decrease in neo-angiogenesis compared with VACV vectors treated alone, followed by reduced tumor volumes, decreased tumor neo-angiogenesis, and increased survival of mice bearing the tumors [128,129,130]. Moreover, when VACV was simultaneously armed with the two anti-angiogenesis factors namely anti-VEGF and anti-EGFR (GLV-1h444), it was found that only minimal tumors were grown after GLV-1h444 injection in mice carrying A549 tumor xenografts, which displayed a better therapeutic efficacy than either of the two single antibodies alone expressing VACV injected ones [131]. Thus, the combination of anti-angiogenesis therapy and oncolytic VACV might be an ideal path to enhance the anti-angiogenesis effects for preventing tumor development.

#### 3.6.3. Combination with Immune Checkpoint Blockade or CAR-T Cell Therapy

Immune checkpoint blockade (ICB) of PD-1/PD-L1 has led to significant tumor regressions and improved survival in a subset of patients with many reported tumor entities [132,133,134,135,136]. Reports showed checkpoint blockage produced rapid and durable immune responses, especially in PD-1-expressing tumor patients [133,137,138]. As a negative signal regulatory protein, PD-1 is expressed on activated T cells to inhibit T cell activation. The ligand PD-L1 is expressed on tumor cells, which binds to T cells expressing PD-1, blocking T cell functions as a tumor evading mechanism, resulting in decreased productions of effector cytokines and lower cytolytic activity [138,139,140]. Therefore, the blockade of the PD-1/PD-L1 pathway can promote the activation of tumor-specific T cells, which leads to enhanced anti-tumor T cell responses [141]. However, the majority of tumor patients are still refractory to immune checkpoint inhibition, which might be because of the lack of pre-existing immune responses [136,141,142], an important condition for successful ICB therapy [143].

OVs infection of tumors might overcome resistance to ICB by improving the magnitude and quality of anti-tumor T cell responses [136,144]. Reports showed dramatic tumor regression and metastatic inhibition after combined usage of anti-PD-1 antibody and oncolytic VACV in a previously treatment-refractory carcinoma model [145], with an increased CD8^+^ T cells infiltration and DC activation [47]. The monotherapy of metastatic sarcoma with ICB showed little activity [145], demonstrating the excellent benefits to enhance T cells activity after oncolytic VACV in combination with ICB. In a pancreatic ductal adenocarcinoma treatment study, oncolytic VACV and anti-PD-L1 combination also displayed good tumor attacking ability [146], and the complementary benefit of oncolytic VACV in ICB treatment was further confirmed. A clinical phase II trial in patients with advanced NSCLC is underway to evaluate the anti-tumor efficacy of the combination therapy with modified VACV and nivolumab, an agent of anti-PD-1 monoclonal antibody (NCT02823990) [138].

Chimeric antigen receptor (CAR) T-cell therapy has been one of the most promising immunotherapies in the past decade. Despite the recent success of CAR T-cell therapy in hematologic malignancies, especially in B-cell lymphoma and acute lymphoblastic leukemia, the application of this treatment approach in solid tumors has faced several obstacles: the heterogeneous expression of antigens and the induction of effective immune responses in the immunosuppressive TME [147,148]. Previous studies showed that VACV (expressing a non-signaling truncated CD19 (CD19t) protein for tumor-selective delivery) enabled targeting by CD19-CAR T cells. OV19t induced local immunity characterized by tumor infiltration of endogenous and adoptively-transferred CAR T cells in several mouse tumor models [149]. A combination immunotherapy approach using oncolytic viruses might promote de novo CAR T cell targeting of solid tumors.

## 4. Precaution and Risk of VACV in Tumor Therapy

The use of viruses to treat tumors is the result of several observations originating in the mid-1880s when leukemia patients would occasionally go into remission upon influenza infection [150]. In the 1950s, virus-based therapies gained momentum and several pre-clinical and clinical trials were done to explore their potential for cancer treatment [2]. The recombinant DNA technology in the early 1990s has enabled the enhancement of both the oncolytic properties and the safety profile of OVs. There might be two aspects for some viruses to proliferate better in tumor cells compared to normal cells, one is that the viruses rely on cellular mechanisms to replicate themselves. Thus, viruses prey upon activated pathways in tumor cells for efficient replication. For example, VACV requires activation of the EGFR pathway to efficiently replicate. Lee et al. found that there were a large number of mutations in the lung tumor cells from a single patient, many of these lead to overlapping, redundant activation of EGFR and parallel pathways [151]. On the other hand, tumor cells have found ways to overcome or attenuate their apoptotic programs and, thus, cannot resist viral infection and spread [152].

VACV has been used as a eukaryotic cloning vector for the expression of heterologous genes shortly after the eradication of smallpox [153]. A highly attenuated VACV strain MVA has been widely used as a vector for developing vaccines for infectious diseases and cancer [154,155]. Diversified recombinant VACVs provide more options for the treatment of different types of tumors. Especially the utilization of neoantigens in VACV vectors, which may make personalized tumor treatment possible. In addition, oncolytic VACV or VACV armed with cytokine/chemokines, or co-stimulating factors can profoundly affect the TME. The TME has emerged as a potential therapeutic target in solid tumors. Varies immune cells can present in TME and suppress ant-tumor immune responses in normal settings [155]. VACV induces strong Th1 immune responses, which help to break the immunosuppressive TME and promote anti-tumor immune responses. A series of oncolytic VACVs for clinical tumor treatments with good safety and efficacy are listed in Table 1. However, a phase 3 randomized, open-label study comparing Pexa Vec (JX-594, Vaccinia GM-CSF/TK-Deactivated Virus), followed by sorafenib versus sorafenib in patients with advanced hepatocellular carcinoma (HCC) without prior systemic therapy (JX594-HEP024), failed to reach objectives. This means that more in-depth mechanisms are needed for the application of VACVs in tumor treatment; Meanwhile, the individual status of tumor patients and the factors that may lead to treatment failure need to be further clarified.

VACV infection can establish long-term immune memory which will result in faster clearance of viruses in subsequent applications, which is a disadvantage of VACV as OV vectors. Using different strains of VACV, combining VACV with other OV vectors or therapeutic approaches might achieve better antitumor effects. In addition, it might be not safe to use VACV in some immunodeficient cancer patients. Individuals with immune deficits might contract severe, even fatal disseminated VACV infection. Certain engineered VACVs may present an even greater risk to such individuals and result in the person-to-person spread of the vaccine strains. The risks of using live-attenuated VACVs for cancer therapy need to be considered and evaluated based on the personal conditions of each cancer patient.

## 5. Summary

VACV is an ideal platform to generate rationally designed OVs. VACV-based OVs provide the unique advantage of targeting and lysing cancer cells and at the same time stimulating strong anti-tumor immune responses. Moreover, the combination of VACVs with other immunotherapeutic approaches, such as ICB, has the potential to improve the outcome in cancer patients. Combining VACV during CAR-T cell therapy might help to improve the efficiency of CAR-T cells in solid tumors. Furthermore, rationally designed OVs will help the field shift towards personalized cancer treatment. Meanwhile, the safety and rational treatment designs of using VACVs in cancer patients need to be comprehensively considered and prepared in advance.

## Figures and Tables

**Figure 1 vaccines-10-01552-f001:**
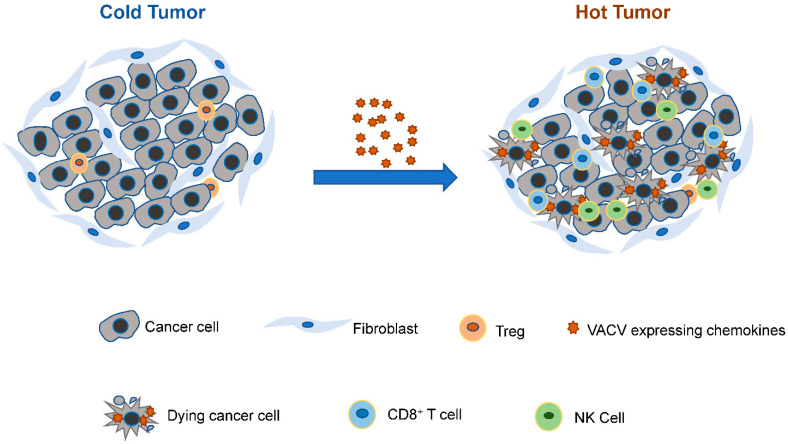
VACV infection might turn the cold tumor into a hot tumor. Recombinant chemokine-expressing VACV infects and kills tumor cells. At the same time, chemokines will be produced, and CD8^+^ T cells and NK cells will be recruited into the tumor, turning the cold tumor into a hot tumor to achieve better therapeutic antitumor efficacy. VACV combined with chemokine therapy may also achieve therapeutic antitumor efficacy.

**Figure 2 vaccines-10-01552-f002:**
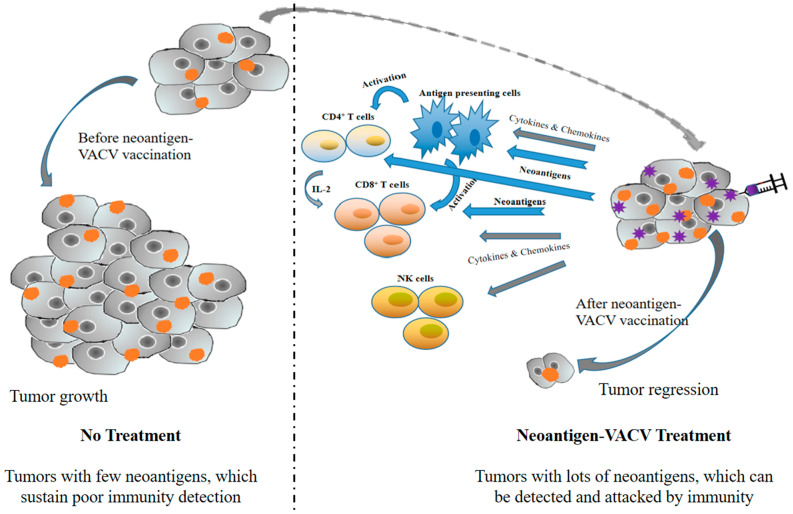
Simplified mechanism of neoantigen-VACV in tumor cells treatment. Before neoantigen-VACV vaccination, tumors with few neoantigens sustain poor immunity detection. However, after neoantigen-armed VACV vaccinations, the neoantigen expressions of tumors are increased with the replications of VACV, so that tumors can be detected and attacked by the immune cells.

**Table 1 vaccines-10-01552-t001:** Representative clinical studies of oncolytic VACV.

Name	Description	Delivery Route	Cancer	Co-Therapy	Phase	Status	Reference
JX-594	Wyeth Strain; Deletion: TK Transgenes: GM-CSF β-galactosidase	i.t. (intrathecal)	Solid tumors	Ipilimumab Tremelimumab	1	Recruiting	NCT02977156
		i.v. (intravenous)	Refractory colorectal cancer	Durvalumab Tremelimumab	1/2	Active, not recruiting	NCT03206073
		i.v., i.t.	Renal cell carcinoma	REGN2810 (Anti-PD-1)	1b	Recruiting	NCT03294083
		i.t.	Hepatocellular carcinoma	Sorafenib	3	Completed	NCT02562755
GL-ONC1 (GLV-1h68)	Lister strain; Deletions: TK (J2R) F14.5L A56R (hemagglutinin genes) Transgenes: GFP β-galactosidase β-glucoronidase	i.v.	Head and neck carcinoma	-	1	Completed	NCT01584284
		i.p. (intraperitoneal)	Ovarian cancer	bevacizumab	1b/2	Recruiting	NCT02759588
		i.p.	Peritoneal carcinomatosis	-	1/2	Completed	NCT01443260
		i.v.	Solid tumors	-	1	Completed	NCT00794131
PROSTVAC-V (Vaccinia-PSA-TRICOM)	Strain: partially attenuated version of the virus used for smallpox immunization [42]; Transgenes: PSA (L155) TRICOM (B7.1, ICAM-1, LFA-3)	s.c. (subcutaneous)	Prostate cancer	(153)Sm-EDTMP (radiation) Sargramostim PROSTVAC-F	2	Completed	NCT00450619
		s.c.	Prostate Cancer	PROSTVAC-F (+/−)GM-CSF	3	Completed	NCT01322490
Vaccinia-CEA-TRICOM	Wyeth strain; Transgenes: CEA TRICOM [43]	s.c.	Breast Cancer	Fowlpox-CEA(6D)/TRICOM Sargramostim Cyclophosphamide Doxorubicin hydrochloride Paclitaxel Radiation	2	Completed	NCT00052351
p53MVA	Ankara strain; Transgenes: Human p53	i.v.	Ovarian cancer Peritoneal cancer Fallopian tube cancer	Pembrolizumab	2	Recruiting	NCT03113487
		s.c.	Colon cancer Gastric cancer Pancreatic cancer Rectal cancer	-	1	Completed	NCT01191684
TG6002	Copenhagen strain; Deletions: TK I4L Transgenes: FCU1	i.v.	Glioblastoma	5-flucytosine (5-FC)	1/2	Recruiting	NCT03294486
		Intrahepatic arterial (IHA) administration	Colorectal cancer	5-FC	1/2	Recruiting	NCT04194034
		i.v.	Gastro-intestinal tumors	5-FC	1/2	Recruiting	NCT03724071
MVA-brachyury- TRICOM	Ankara strain; Transgenes: Brachyury TRICOM	s.c.	Lung cancer Breast cancer Prostate cancer Ovarian Cancer	-	1	Completed	NCT02179515
TG4010	Ankara strain; Transgenes: MUC1 Human IL-2	s.c.	Non-small cell lung cancer	Nivolumab	2	Active, not recruiting	NCT02823990
		-	MUC-1 positive advanced cancer	-		Completed	NCT00004881
		s.c.	Non-small cell lung cancer	Chemotherapy Nivolumab	2	Active, not recruiting	NCT03353675
vvDD-CDSR	Western Reserve strain; Deletion: TK VGF Transgenes: LacZ Cytosine deaminase and somatostatin receptor genes [44]	i.t. i.v.	Melanoma Breast cancer Head and neck squamous cell cancer Liver cancer Colorectal cancer Pancreatic adenocarcinoma	-	1	Completed	NCT00574977
MVA-5T4 (TroVax^®®^)	Ankara strain; Transgenes: Tumor antigen 5T4	i.m. (intramuscular)	Ovarian cancer Fallopian tube cancer Peritoneal cancer	-	2	Completed	NCT01556841

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
