# Peer review of "Design Strategies and Precautions for Using Vaccinia Virus in Tumor Virotherapy"

_vaccines, 2022, doi:10.3390/vaccines10091552_

Round 1
Reviewer 1 Report
The manuscript "Oncolytic Vaccinia Virus in Tumor Treatment" provides comprehensive analytical review of the literature on the vaccinia virus and its derivatives for antitumor therapy. Most of the issues concerning the properties of oncolytic viruses are covered and explained. The manuscript should be of interest to the readers of Cancer Vaccines and Immunotherapy in its present form. However resolving some minor points could improve the manuscript. There is a long story of discovering an oncolytic activity of some viruses (including vaccinia virus). However the question, how do virus "knows" that it is oncolytic against certain tumor (in other words - how do it acquired increased potential to proliferate in tumor cells compared to the normal cells), remains poorly discussed. Some speculations on this topic from the authors might greatly improve the value of the manuscript. The article is written in a good understandable language. However some statements are not completely clear. For example the one on line 234-235.
Author Response
We thank the reviewer for the helpful comments. There might be two aspects for some viruses to proliferate better in tumor cells compared to the normal cells, one is that the virus prey upon activated pathways in tumor cells for efficient replication, the other is that tumor cells have found ways to overcome or attenuate their apoptotic programs, thus cannot resist viral infection and spread. We added some speculations on this topic in the revised manuscript. We checked the manuscript carefully and revised some unclear sentences in the revised manuscript.
Reviewer 2 Report
Liu et al present a review addressing the potential of vaccinia virus in cancer treatment. This is a extensive review, however some concerns need to be addressed before publication.
- The manuscript needs extensive grammar review. Several points where the subject does not match the verb; extremely long sentences that are hard to understand (line 184, 309); sentences that start and end with the same conclusion (line 154 for example).
- The authors describe several examples of the use of VACV in tumors, however often the tumor type and if in vivo or in vitro models are used are missing (for example: lines 115; 139, 192, 270, 348)
- Table 1 should be more succinct (use bullet points, instead of full sentences), a lot of the information is repeated in the text
- The authors refer CAR-T cell therapy on the summary, but this therapeutic approach was never mentioned on the text.
Author Response
We would first like to thank the reviewer for the helpful comments. Reviewers’ questions are in bold and our reply in regular type.
Liu et al present a review addressing the potential of vaccinia virus in cancer treatment. This is a extensive review, however some concerns need to be addressed before publication.
- The manuscript needs extensive grammar review. Several points where the subject does not match the verb; extremely long sentences that are hard to understand (line 184, 309); sentences that start and end with the same conclusion (line 154 for example).
We thank the reviewer for careful reading. We checked the manuscript carefully and corrected several grammar mistakes and rewrote some extremely long sentences in the revised manuscript.
- The authors describe several examples of the use of VACV in tumors, however often the tumor type and if in vivo or in vitro models are used are missing (for example: lines 115; 139, 192, 270, 348)
We thank the reviewer for the suggestion. The tumor type or tumor models were added in the revised manuscript.
- Table 1 should be more succinct (use bullet points, instead of full sentences), a lot of the information is repeated in the text
We revised Table 1 succinctly as suggested by the reviewer.
- The authors refer CAR-T cell therapy on the summary, but this therapeutic approach was never mentioned on the text.
We thank the reviewer for the question. We added some discussion of combination with CAR-T cell therapy in the revised manuscript.
Reviewer 3 Report
The review entitled “Oncolytic Vaccinia Virus in Tumor Treatment” by Xinjun Liu and co-authors is an attempt at summarizing of all the possible uses of vaccinia virus in tumor immunity. Vaccinia virus can be readily engineered to express a variety of products that can impact immune function and potentially alter tropism for tumor cells. The lytic properties of vaccinia virus for infected cells, including some evidence of greater replication in tumor cell lines, make it a prime vector for oncolytic approaches. A wide variety of vaccinia viruses expressing a myriad of products that may impact immunity, either directly or indirectly, have been engineered and tested in models. However, this review fails to note that the initial Phase III trial of a vaccinia virus expressing GM-CSF has failed to reach objectives in hepatocellular carcinoma which casts doubt on its prospects for other cancers. Moreover, vaccinia viruses are not entirely benevolent as individuals with immune deficits can contract severe, even fatal disseminated infection. Certain engineered vaccinia viruses may present an even greater risk to such individuals and person-to-person spread of vaccine strains has been documented. The risks of using live-attenuated vaccinia virus vaccines for cancer therapy may be outweighed by the rewards but both need to be considered. This manuscript does not debate this issue but instead presents a highly speculative treatise on the many potential uses of vaccinia viruses in cancer therapy largely based on other reviews of literature relevant to other oncolytic viruses and immunomodulatory approaches in cancer. Many comments are not appropriately referenced. In summary, the focus of this review on the oncolytic properties of vaccinia is unclear and there are many recent reviews that present a more cogent summary of the field.
Author Response
We thank the reviewer for the insightful comments. The focus of our manuscript is on the design strategies and approaches which might make the vaccinia virus (or other oncolytic viruses) more practical for tumor treatment. We thank the reviewer for pointing out the failure of Phase III trial of a vaccinia virus expressing GM-CSF, we missed this information during our initial references research, we added discussion about this in the revised manuscript. We discussed a little bit about the potential risks of using live-attenuated vaccinia virus in certain patients, we extend the discussion the revised manuscript. We carefully checked the references and corrected several mistakes.
Round 2
Reviewer 2 Report
The authors addressed most of the comments.
However, table 1 is still too complex as is. Not much as changed, except the adding of bullet points. Table 1 should be significantly simplified (should look like table 2). Otherwise there is no point on having the table if the information is convened on the text.
Author Response
Point by point response to reviewers
We would first like to thank the reviewers for their helpful comments. Reviewers’ questions are in bold and our reply in regular type.
Comments and Suggestions for Authors
The authors addressed most of the comments.
We thank the reviewer for the supportive comments on our revision.
However, table 1 is still too complex as is. Not much as changed, except the adding of bullet points. Table 1 should be significantly simplified (should look like table 2). Otherwise there is no point on having the table if the information is convened on the text.
We agree with the reviewer that table 1 is still too complex and the information is all convened on the text. We deleted table 1 to make the revised manuscript more concise and focused. We thank the reviewer for the suggestion.
Reviewer 3 Report
This revised manuscript reviews and speculates about the use of oncolytic vaccinia viruses in cancer therapy. Reasonably comprehensive with respect to published studies a number of unsubstantiated comments that would support the use of these viruses detract from the review aspects of this manuscript. For example it is stated that the use of vaccinia viruses to express tumor associated antigens would bypass central tolerance to these antigens which is incorrect. The risk/reward issue of using live attenuated viruses to kill tumor cells in people who may have immune deficits due to their disease or prior treatments is very much underplayed and should be a major element of this kind of review. There is an extensive history of the used of these viruses to express other antigens for use in vaccination which provides some insight into risk/reward in healthy individuals but considerably less is known about the risk in cancer patients. In addition, the review tends to mix up vaccinia virus use as an oncolytic virus versus its considerable more extensive use as a vector to express tumor antigens and other immunomodulators. Organization of the manuscript into clearly defined section directed at the various topics would help.
Author Response
Point by point response to reviewers
We would first like to thank the reviewers for their helpful comments. Reviewers’ questions are in bold and our reply in regular type.
Comments and Suggestions for Authors
This revised manuscript reviews and speculates about the use of oncolytic vaccinia viruses in cancer therapy. Reasonably comprehensive with respect to published studies a number of unsubstantiated comments that would support the use of these viruses detract from the review aspects of this manuscript. For example it is stated that the use of vaccinia viruses to express tumor associated antigens would bypass central tolerance to these antigens which is incorrect. The risk/reward issue of using live attenuated viruses to kill tumor cells in people who may have immune deficits due to their disease or prior treatments is very much underplayed and should be a major element of this kind of review. There is an extensive history of the used of these viruses to express other antigens for use in vaccination which provides some insight into risk/reward in healthy individuals but considerably less is known about the risk in cancer patients. In addition, the review tends to mix up vaccinia virus use as an oncolytic virus versus its considerable more extensive use as a vector to express tumor antigens and other immunomodulators. Organization of the manuscript into clearly defined section directed at the various topics would help.
We thank the reviewer for the insightful comments. In theory, TSAs are antigens specifically expressed on tumor cells and not on normal cells. Thus, TSA might be able to stimulate TSA-specific T cell responses. We agree with the reviewer that it is not rigorous to state that TSA would bypass central T cell tolerance. This inaccurate statement was in table 1 in the original manuscript, we deleted table 1 in the revised manuscript to make it more concise and focused. We agree with the reviewer that using viruses to express other antigens for use in vaccination provides some insight into risk/reward in healthy individuals, we thank the reviewer for the suggestion. We discussed the long use of viruses as vectors to express other antigens in the revised manuscript. We reorganized the revised manuscript into clearly defined sections as suggested by the reviewer.
Round 3
Reviewer 2 Report
The authors addressed the comments.
Author Response
Point by point response to reviewers
We would first like to thank the reviewers for their helpful comments. Reviewers’ questions are in bold and our reply in regular type.
Comments and Suggestions for Authors
The authors addressed the comments.
We thank the reviewer for the supportive comments on our revision.
Reviewer 3 Report
This manuscript is fairly comprehensive with respect to the use of Vaccinia virus in tumor therapy such that the title is somewhat misleading; the manuscript encompasses considerably more than the oncolytic aspects of the virus. A variety of the uses of VV as a delivery vector for cytokines, antigens, etc are discussed. A number of issues with the tenses of statements make the manuscript difficult to read. A number of statements, such as "In addition, GM-CSF stimulates a long-lasting and effective anti-tumor immunity by priming CD4+ and CD8+ T cells[36]." which erroneously implies that GM-CSF "primes" T cells, need to be corrected.
Author Response
Point by point response to reviewers
We would first like to thank the reviewers for their helpful comments. Reviewers’ questions are in bold and our reply in regular type.
Comments and Suggestions for Authors
This manuscript is fairly comprehensive with respect to the use of Vaccinia virus in tumor therapy such that the title is somewhat misleading; the manuscript encompasses considerably more than the oncolytic aspects of the virus. A variety of the uses of VV as a delivery vector for cytokines, antigens, etc are discussed. A number of issues with the tenses of statements make the manuscript difficult to read. A number of statements, such as "In addition, GM-CSF stimulates a long-lasting and effective anti-tumor immunity by priming CD4+ and CD8+ T cells[36]." which erroneously implies that GM-CSF "primes" T cells, need to be corrected.
We thank the reviewer for the insightful comments. We agree with the reviewer that the current title is too broad, as the manuscript is mainly focused on the design strategies and approaches which might make the vaccinia virus (or other oncolytic viruses) more practical for tumor treatment. We changed the manuscript title to: “Design strategies and precautions for using vaccinia virus in tumor virotherapy” in the revised manuscript. We think that the new title is more specific and suitable for the content of the manuscript. We hope that the reviewer and editor would agree with us, and approve the new title. We thank the reviewer for the careful reading, we carefully checked the manuscript and corrected several exaggerated or inaccurate comments, especially the statements about GM-CSF.